# Crohn’s Disease, Host–Microbiota Interactions, and Immunonutrition: Dietary Strategies Targeting Gut Microbiome as Novel Therapeutic Approaches

**DOI:** 10.3390/ijms23158361

**Published:** 2022-07-28

**Authors:** María A. Núñez-Sánchez, Silvia Melgar, Keith O’Donoghue, María A. Martínez-Sánchez, Virgina E. Fernández-Ruiz, Mercedes Ferrer-Gómez, Antonio J. Ruiz-Alcaraz, Bruno Ramos-Molina

**Affiliations:** 1Obesity and Metabolism Research Laboratory, Biomedical Research Institute of Murcia (IMIB), 30120 Murcia, Spain; mariaa.nunez@imib.es (M.A.N.-S.); mariaantonia.martinez1@um.es (M.A.M.-S.); virginiaesperanza.fernandez@um.es (V.E.F.-R.); bruno.ramos@imib.es (B.R.-M.); 2APC Microbiome Ireland, University College Cork, T12 YT20 Cork, Ireland; s.melgar@ucc.ie (S.M.); kwodonoghue@gmail.com (K.O.); 3Department of Endocrinology and Nutrition, Virgen de la Arrixaca University Hospital, 30120 Murcia, Spain; 4Department of Biochemistry and Molecular Biology B and Immunology, Faculty of Medicine, University of Murcia, 30120 Murcia, Spain

**Keywords:** Crohn’s disease, gut microbiome, diet, immunonutrition, inflammation

## Abstract

Crohn’s disease (CD) is a complex, disabling, idiopathic, progressive, and destructive disorder with an unknown etiology. The pathogenesis of CD is multifactorial and involves the interplay between host genetics, and environmental factors, resulting in an aberrant immune response leading to intestinal inflammation. Due to the high morbidity and long-term management of CD, the development of non-pharmacological approaches to mitigate the severity of CD has recently attracted great attention. The gut microbiota has been recognized as an important player in the development of CD, and general alterations in the gut microbiome have been established in these patients. Thus, the gut microbiome has emerged as a pre-eminent target for potential new treatments in CD. Epidemiological and interventional studies have demonstrated that diet could impact the gut microbiome in terms of composition and functionality. However, how specific dietary strategies could modulate the gut microbiota composition and how this would impact host–microbe interactions in CD are still unclear. In this review, we discuss the most recent knowledge on host–microbe interactions and their involvement in CD pathogenesis and severity, and we highlight the most up-to-date information on gut microbiota modulation through nutritional strategies, focusing on the role of the microbiota in gut inflammation and immunity.

## 1. Introduction

Crohn’s disease (CD), a subtype of inflammatory bowel disease (IBD), refers to a complex disabling, idiopathic, progressive, and destructive disorder with an unknown etiology that could affect any segment of the gastrointestinal (GI) tract [1]. It is estimated that CD could affect up to 300 in 100,000 individuals in westernized countries in Europe, North America, and Oceania, and it is associated with high morbidity and a high economic burden [2]. Crohn’s disease is a chronic remitting and relapsing inflammatory disease characterized by skip intestinal lesions affecting the gut wall along the GI tract, which can lead to chronic abdominal pain, diarrhea, obstruction, and/or perianal lesions [3]. The pathogenesis of CD is multifactorial and involves the interplay between the host’s genetics, immune system, and gut microbiota, which are influenced by environmental factors and result in an aberrant response in the GI tract with subsequent intestinal inflammation [4].

There are a variety of pharmacological treatment options for patients with CD, including antibiotics, 5-aminosalicylates, corticosteroids, immune suppressants, and/or biologic therapy, all of which are principally selected based on the symptoms and whether remission is being induced or maintained. Although traditionally corticosteroids have been the cornerstone of CD management, anti-inflammatory therapies (e.g., anti-tumor necrosis factor (anti-TNF) therapy) have become the therapy of choice for the treatment of CD in the past two decades, especially in patients with moderate to severe active CD or in patients that are unresponsive to other therapies [5]. In addition, new biologic therapies including anti-integrins and anti-IL12/23p40 are gaining interest in light of their promising results [6,7]. However, most of these therapies show poor long-term maintenance of intestinal integrity, and they are associated with significant health-care costs and side effects. Thus, the development of new strategies such as nutritional interventions is mandatory to improve the quality of life of patients with CD [1]. In this sense, exclusive enteral nutrition (EEN) has been successfully used over a long period of time in the treatment of patients with CD, and it is recommended as the first-line induction therapy for children with CD [8]. Indeed, EEN has been demonstrated to achieve similar results to those obtained with corticosteroid treatment in the patients, with additional benefits such as avoidance of growth retarding and complete coverage of nutritional needs [8,9]. However, although the efficacy of EEN in the clinical management of CD is indubitable, the exact mechanisms underlying such positive outcomes remain uncertain [10].

The intestinal gut microbiome has emerged as a pre-eminent target for potential new therapeutic treatments in CD. The human gut microbiota harbors more than 10^14^ microorganisms (including bacteria, virus, and yeast), which have a symbiotic and mutualistic relationship with the host. Since the moment of birth, the gut microbiota plays an important role in physiological processes such as the development of the immune system, intestinal homeostasis, behavior, and host metabolism [11,12,13]. Imbalance in the gut microbiome, the so-called dysbiosis, is associated with metabolic and gastrointestinal conditions such as IBD, which includes CD, and ulcerative colitis (UC) [14]. Among the factors that meaningfully affect the microbiota, diet is one of the key players in maintaining a well-balanced and healthy gut-microbial microenvironment. Thus, the development of specific diets aimed at the modification of gut microbiota has arisen as a promising cost-effective strategy to improve CD management and evolution. In this review, we discuss the most recent knowledge on host–microbe interactions and their involvement in the pathogenesis and severity of CD, and we highlight the most up-to-date information on nutritional interventions targeting the modulation of gut-microbiota composition and the immune system.

## 2. Gut Microbiome in Crohn’s Disease

The gut microbiome is closely linked to the immune system and is a key player in the pathogenesis of CD [15]. The emergence of new molecular techniques and bioinformatics tools in the last decade has provided a better understanding of the alterations associated with a pathological status of the microbiome and its associated metabolome. Multiple human microbiome studies have demonstrated a close relationship between such dysbiosis and certain clinical aspects of CD, including inflammation, intestinal permeability, and postoperative CD recurrence [16,17]. Thus, these studies have established that patients with CD showed reductions in bacterial diversity and altered abundance of certain taxa including a reduction in health-promoting microorganisms (e.g., *Faecalibacterium* and *Roseburia* spp.) and an increase in pathogenic microorganisms (e.g., *Escherichia*, *Fusobacterium*, and *Mycobacterium* spp.) [18]. Furthermore, increasing evidence support that such dysbiosis might be a causal factor in the development and evolution of chronic intestinal inflammatory diseases such as CD [19]. Despite this growing evidence, the exact mechanisms involved in host–microbe interactions in CD pathophysiology have not been yet fully understood.

### 2.1. Intestinal Inflammation and Gut Dysbiosis in CD

Susceptibility to CD is dependent on different elements, including genetic predisposition and environmental factors, such as diet and pollution. Importantly, these non-genetic factors are well-known to have an important impact on the gut microbiome composition of the host [20]. Microbiome components interact with the host’s immune system to play a key role in the maintenance of physiological homeostasis. Thus, the disruption of a healthy microbiome or the reduction in the ratio of certain beneficial commensal microorganisms, resulting in a dysbiosis, may induce an exacerbated activation of the mucosal immune system associated with an exacerbated and altered cytokine production that contributes to the establishment and progression of CD [21].

Several pro-inflammatory cytokines are involved in CD, but, among them, IL-23 [22] (a member of the IL-12 family) and IL-17 are key to its pathogenesis [23]. These pro-inflammatory cytokines activate and expand a lymphocyte T helper (Th) 17 response that is accompanied by the induction of other pro-inflammatory mediators including TNFα, IFNγ, and IL-1β, among others. Gut-resident macrophages, together with dendritic cells, play a key role in the establishment of this exacerbated pro-inflammatory process found in CD. These cells are the main producers of IL-23 [24], which activates the Th17 and Th1 inflammatory response, but they are also involved in the progression of CD as a later cellular source of other pro-inflammatory cytokines, such as TNFα and IL-1β. The other cellular key players in the progression of the pathology are Th17 lymphocytes, which expand and augment their population in the gut mucosa in response to IL-23 [25], maintaining chronic inflammation of the intestine in CD. Th17 lymphocytes mainly produce IL-17, but can also produce IFNγ, IL-21, IL-22, and TNFα [26,27]. Thus, the accumulation of active expanding Th17 lymphocytes in the submucosa and lamina propria further contributes to the progression of CD [28]. Other cell populations, such as certain subsets of γδ T cells [29], natural killer T (NKT) cells [30,31,32], and type 3 innate lymphoid cells (ILC-3) [22], also respond to IL-23 and other pro-inflammatory cytokines and are, thus, considered as “type 17 cells”. The stimulation of these type 17 cells with IL-23 and other pro-inflammatory mediators, such as IL-1β, also contribute to the local mucosa inflammation, fulfilling an important role in CD perpetuation [33]. As a counterpart, regulatory T cells (Tregs) have an important role as main down-regulators and major suppressors of the immune response [34], and, thus, the differentiation of T cells to this modulatory phenotype and their appropriate activation may fulfil a key role in the maintenance of the gut homeostasis.

On the other hand, it has been reported that the gut dysbiosis observed in IBD patients [35] is characterized by a reduction in microorganisms with anti-inflammatory properties and an elevation of those with pro-inflammatory capacities [36,37], which favors the augmented and disturbed pro-inflammatory cytokine production observed in CD. A reduction in the general diversity of the gut microbiota, accompanied by a lower abundance of Firmicutes, is a common signature of IBD [38,39]. In patients with CD, it has been shown that the abundances of *Faecalibacterium prausnitzii*, *Blautia faecis*, *Roseburia inulinivorans*, *Ruminococcus torques*, and *Clostridium lavalense* are highly reduced when compared to healthy/control individuals. Notably, *F. prausnitzii* presents an important anti-inflammatory activity mediated by its butyrate production [40,41]. Indeed, these bacteria have been demonstrated to induce the production of the anti-inflammatory cytokine IL-10 by immune cells, thus being able to diminish the production of key pro-inflammatory cytokines such as IL-12 and IFNγ [42]. Furthermore, short-chain fatty acid (SCFA)-producing bacteria strains present in healthy human fecal samples, such as those included in *Clostridium* clusters IV, XIVa, and XVIII, can induce, via butyrate production, the differentiation and expansion of anti-inflammatory Tregs [43]. On the contrary, an increase in Proteobacteria, especially those with adhesive properties to the intestinal epithelium such as adhesion-invasive *Escherichia coli* (AIEC), prevalent in CD [44], can induce Th17 pro-inflammatory cells [45].

An open question in this field is whether the dysbiosis seen in CD precedes inflammation or whether dysbiosis is a consequence of the inflammatory process. An interesting study that analyzed the microbiota composition in mucosal tissue biopsies and fecal samples of treatment-naive pediatric patients with CD revealed an increased abundance of *Veillonellaceae*, *Paturellaceae*, *Neisseriaceae*, *Fusobacteriaceae* spp., and *E. coli* spp., and a decreased abundance of *Clostridiales*, *Bacteroides*, *Faecalibacterium* spp., *Roseburia* spp., *Blautia* spp., *Ruminococcus* spp., and *Lachnospiraceae* spp. [18]. As this study investigated a newly diagnosed population, it suggests that microbiota changes occur early and may precede clinical disease. Indeed, a recent study using a genetic model of CD (deficient in two CD susceptibility genes, *NOD2* and phagocyte NADPH oxidase) demonstrated an increase in pathobiont species preceding the onset of colitis [46]. These observations suggest that dysbiosis in CD can be present before inflammation, suggesting a key role of the gut microbiome in CD pathogenesis. Figure 1 shows an overview of the inflammatory mechanisms described in CD in relation to microbial dysbiosis.

### 2.2. Host–Microbe Interactions in CD

Involvement of the gut microbiome in CD pathogenesis was initially suggested by Rutgeerts et al., after observing that CD recurrence was decreased or eliminated in patients undergoing surgical diversion of the fecal stream [47]. These observations were later confirmed by another group, after seeing that exposure of distal limb to luminal content was associated with recurrence of inflammation after surgical resection [48]. Further evidence indicating a key role of microbial involvement in CD comes from animal models, where the transfer of fecal microbiota from mice with colitis-initiated inflammation in healthy mice [49]. Moreover, colitis-susceptible mice with T cell receptor-alpha beta (TCRαβ) mutations develop colitis when colonized with a conventional microbiota but not when raised in germ-free conditions [50].

Many studies have reported changes in the microbiota of patients with IBD patients. However, these have not identified a consistent change in microbial composition. A potential reason for this is the great variation in studies characterizing the microbiota in IBD patients due to confounding variables such as disease duration, differences in treatment, sampling location, and variation in analysis. However, it is very well-characterized that dysbiosis is present in IBD patients, which is more pronounced in CD than in UC, with a more altered and unstable microbiota composition in the former [51,52]. A number of studies have demonstrated that the microbiota of patients with CD possesses a reduced richness of species, with a decrease in the relative abundance of *F. prausnitzii*, *Bacteroides*, *Blautia*, *Ruminococcus*, *Roseburia*, *Coprococcus*, and *Lachnospiraceae*, and increased abundance of *Enterobacteriaceae*, *Fusobacteriaceae*, and *Streptococcaceae* [53,54]. Among bacteria commonly associated with IBD, adherent-invasive *E. coli* (AIEC), initially isolated from patients with CD with ileal lesions [44], has been highly associated with CD pathogenesis [55]. In this regard, a recent study found an association of AIEC with the early phase of recurrence in patients with ileal CD [56]. Additionally, a systemic review and meta-analysis reported that the prevalence of AIEC in patients with UC is 12% (range 0% to 10%) compared to 5% in non-IBD controls and 29% (range 21.7 to 62.5%) in patients with CD, indicating that AIEC may also be relevant in the pathogenesis of UC [57]. AIEC lack classical pathogenicity genes but can persist in macrophages, where they can induce proinflammatory cytokine secretion without inducing cell death [57]. One gene that supports the survival of AIEC in macrophages is *gipA*, which is induced by different factors including bile salts, reactive oxygen species, and pH changes [58].

In addition to infecting macrophages, AIEC can adhere to and invade epithelial cells, thus affecting the integrity of the epithelial barrier by altering the expression of tight junction proteins such as claudin-2, zonula occludens 1, and E-cadherin [59,60]. AIEC can bind to epithelial cells through the receptor carcinoembryonic antigen-related cell adhesion molecule 6 (CEACAM6), often found increased in the ileum of patients with CD [61,62]. Several factors can affect the expression of AIEC on epithelial cells, including pro-inflammatory cytokines, dietary emulsifiers, gut metabolites, etc. [63,64,65]. For example, the fucose fermentation product 1,2-propanediol, which is highly increased in the microbiome of patients with CD, was shown to regulate AIEC-induced intestinal T cell inflammation in mice via the metabolic recognition and activation of phagocytes [64]. Other mucosal metabolites such as ethanolamine, ileitis-associated amino acids, glutathione, and fucose were shown to enhance AIEC growth and virulence factors resulting in worsening intestinal inflammation in AIEC-mono-associated IL-10^−/−^ mice [65], indicating the impact that the gut environment has on the ability for AIEC to thrive in the gut of the patients. Other studies also associated the impaired autophagy-mediated clearance of AIEC with enhanced inflammation, further supporting their role in CD pathogenesis [66].

Similar to alterations in gut microbiota, gut mucosal virome analyses in IBD patients have identified an increase in Caudiovirales phages and viral-like particles (VLP) such as *Siphoviridae*, *Myoviridae*, and *Podoviridae*, especially associated to patients with CD [67,68]. To examine how the innate immune system responds to viruses in the gut, a recent study utilizing humanized mice treated with colonic virome isolated from healthy individuals, showed that the mice were protected from the development of intestinal inflammation, while those treated with UC- and CD-associated viromes developed a more severe colitis phenotype [69]. Interestingly, macrophages cultured in the presence of healthy colonic virome resulted in the down-regulation of genes associated with apoptosis, inflammation, and the anti-viral response and the up-regulation of genes associated with pro-survival and homeostatic/resolving state, while UC/CD virome induced a pro-inflammatory gene profile, often associated to IBD [69]. Moreover, epithelial cells cultured in the presence of UC/CD virome presented TLR4-independent barrier integrity and pro-inflammatory cytokine response. Interestingly, a worsening in the epithelial cell response to IBD virome was observed when mutations associated with the IBD-susceptible gene melanoma differentiation-associated gene 5 (MDA5) were present [69].

Fungi are also important microorganisms constituting the gut microbiota. Fungi can be recognized by the immune system through several pattern recognition receptors (PRRs) including Toll-like receptors (TLRs), C-type lectin receptors (CLRs), and NOD-like receptors (NLRs). Recognition of the fungal structures, including polysaccharides (mannans or mannoproteins), β-glucans, and unmethylated DNA, by different PRRs, results in the activation of pro-inflammatory cytokines such as IL-1β, TNF-α, etc., leading to an enhanced immune response [70]. Fungal alterations have been identified in patients with IBD, especially in *Candida albicans* and *Malassezia restricta* [71]. Colonization of mice with these fungal species worsened colitis [72,73], and *M. restricta* was shown to activate the NLRP3 inflammasome via an increased caspase-1 and IL-1β activity [73,74]. A higher relative abundance of colonic *M. restricta* was associated with a mutation in the *CARD9* gene, *CARDS12N*, which is linked to IBD onset [73]. In another study, Ost et al. showed that fecal IgA from patients with IBD bound with high affinity to *C. albicans* and low affinity to *Saccharomyces cerevisiae*. In addition, the hyphae of *C. albicans* and an IgA-targeted adhesin led to the exacerbation of intestinal inflammation, and vaccines that induced an adhesin-specific immune response protected the animals from disease [75].

Other data has also indicated that the interaction of fungal and bacterial microorganisms could regulate the outcome of IBD. Thus, a reduction in the abundance of *Enterobacteriaceae* caused by the presence of *C. albicans* reduced the development of murine colitis [76]. Overall, all these recent studies indicate that not only bacteria have an impact on the host response and CD pathogenesis, but also other microorganisms, including fungi and virome, are key players in CD pathogenesis.

### 2.3. Impact of Microbial-Derived Metabolites in CD Pathogenesis

As described above, compositional changes in the gut microbiota are potential contributing factors in driving inflammation in CD. Importantly, alterations in the gut bacteria will alter the bacterially generated metabolite landscape of the gut. These bacterial metabolites (e.g., secondary bile acids (BAs), SCFAs, etc.), have been shown to have an impact on many host processes including metabolism, epithelial barrier integrity, and innate and adaptive immune responses [77].

Numerous reports have demonstrated that BAs can act as signaling molecules, influencing multiple metabolic pathways [78,79,80]. These molecules are synthesized in the liver and secreted via the gall bladder to the small intestine as primary BAs, thus representing a small portion of BA moieties present in the BA pool. In the GI tract, the high diversity of BA moieties is a consequence of the microbial metabolism, which is mainly involved in the conversion of primary BAs into secondary BAs. Therefore, the relative composition of this BA pool is dependent on the gut-microbiota composition, and alterations of this pool have been related to alterations in host signaling pathways both in the gut and systemically. Remarkably, CD pathogenesis has been related to an imbalance in the primary/secondary BA ratio and altered BA concentrations, as well as to impaired BA metabolism (e.g., decreased BA deconjugation) [81]. Furthermore, changes in the BA pool have been proposed as an indicator of therapeutic response, where patients with CD with increased serum levels of secondary bile acids such as deoxycholic acid (DCA) responded better to infliximab, while those with increased levels of unconjugated cholic acid (CA) and chenodeoxycholic acid (CDCA) did not respond [82]. Recent reports have postulated an anti-inflammatory potential of certain BA intermediate moieties, including the lithocholic acid (LCA) metabolites 3-oxo-LCA and isoalloLCA, which inhibit Th17 differentiation, and how bacteria that possess 3α-hydroxysteroid dehydrogenase—the enzyme that catalyzes the production these metabolites—are significantly decreased in IBD patients [83].

There is a growing body of evidence suggesting that BA signaling through BA receptors (BARs), including farnesoid X receptor (FXR), a master regulator of BA synthesis [84], Takeda-G-protein-receptor-5 (TGR5) [85], and the vitamin D receptor (VDR) [86], can influence immune processes [87]. Indeed, activation of these BARs has been shown to exert anti-inflammatory effects through different mechanisms, including the modulation of inflammatory pathways such as NF-κB, the induction of Treg cell differentiation [86,88], the reduction in the levels of pro-inflammatory cytokines such as IL-1β, IL-6, IFNγ, and TNFα [89], and the increased release of the anti-inflammatory cytokine IL-10 [85,90].

SCFAs are bacterial metabolites derived from the fermentation of indigestible fibers, of which acetate, propionate, and butyrate are the most abundantly produced by the gut microbiota [91]. The main functions of SCFAs in the intestine consist of maintenance of homeostasis, intestinal epithelial-cell turnover, energy metabolism, or induction of epithelial-barrier function [92]. In addition, SCFAs can act as an energy source for colonocytes as well as exert an immunomodulatory effect. In particular, butyrate has been widely described to promote an anti-inflammatory response, through the differentiation of Treg cells as well as through the inhibition of NF-κB signaling and the activity of histone deacetylases (HDACs) [93,94], via its interaction with G-protein coupled receptors (GPCRs) (e.g., GPR41, GPR43, and GPR109A). The main butyrate-producing bacteria include the *Roseburia* and *Faecalibacterium* genera, belonging to the Firmicutes phylum [95,96,97], which are known to be significantly reduced in patients with CD [92]. Consequently, levels of luminal butyrate are diminished in these patients, leading, thus, to an exacerbated immune response [98,99]. The effects of SCFA in IBD pathogenesis have been widely studied and their role as inflammation regulators has been recently reviewed by others [100,101].

### 2.4. Other Microbial-Derived Components Related to CD

Outer Membrane Vesicles (OMVs), which are small, spherically bilayer (100–300 nm) vesicles generated by Gram-negative bacteria have been recently described to play a role in the pathophysiology of IBD [102]. In epithelial cells, OMVs specifically secreted by AIEC stimulated IL-8 secretion and promoted AIEC internalization into the mucosa [103,104]. OMVs produced from another IBD-associated pathobiont, *Bacteroides vulgatus*, were reported to both silence dendritic cells [105], activate NF-κB, and stimulate IL-8 production in epithelial cells [106]. A similar immunomodulatory potential has been ascribed to *B. fragilis* OMVs containing polysaccharide A (PSA), which can regulate TLR4 transcription in epithelial cells [107] and increase the production of the anti-inflammatory cytokine IL-10 [108]. Another recent study reported that *B. thetaiotaomicron* OMVs stimulated the expression of IL-10 in colonic dendritic cells, as well as IL-10 and IL-6 in blood-derived dendritic cells in healthy individuals, but not in colonic or peripheral dendritic cells in patients with either CD or UC [109]. Overall, these data indicate an immunomodulatory potential of OMVs by targeting mucosal and systemic cell responses, which are highly dependent on the target cell.

## 3. Nutritional Strategies in CD Treatment and Management

### 3.1. Microbiota-Based Therapies

As previously stated, the gut microbiota has been recently recognized as one of the main factors involved in the pathogenesis of CD. Studies on the microbiome have shown that patients with CD have dysbiosis with decreased diversity, high instability, and high inter-individual variability [110,111]. Furthermore, a reduction in Firmicutes and Actinobacteria, together with an increase in Proteobacteria, are common hallmarks in patients with CD [52]. Thus, it is not surprising that over the last decades, alternative strategies based on the use of pre-/probiotics, have been developed to complement or even replace pharmacological therapy for the treatment of IBD [112]. The role of prebiotics and probiotics in CD management has been extensively discussed and reviewed in detail elsewhere [113,114,115,116,117]. Several probiotic strains known to have beneficial effects on health have been tested in human clinical studies including *Bifidobacterium* spp., *Lactobacillus* spp., *E. coli* Nissle 1917, and *Saccharomyces boulardii* [19]. Noteworthy, while the efficacy of probiotics such as VSL#3, containing a mixture of eight bacterial strains including four Lactobacillus spp., three strains of Bifidobacterium spp., and *Streptococcus salivarius* subsp. Thermophilus, has been well established for UC management, but attempts to prove their usefulness in CD have produced controversial results [113]. These incongruences might be in part explained by differences in study design, methodology, the variety of pre- or probiotics used, and/or patients’ compliance. Thus, further research is needed to confirm the above-mentioned health claims [118]. Due to this controversy, the European Society for Clinical Nutrition and Metabolism (ESPEN) guidelines on clinical nutrition in IBD do not currently recommend (and even discourage) the use of probiotics for the treatment and management of CD [9].

Alternatively, prebiotics (non-digestible food ingredients that stimulate the growth of beneficial bacteria), such as oligosaccharides, inulin, or polyphenols, have been proposed as an option for the modulation of the gut microbiome as CD therapy. For instance, inulin supplementation in a rat model of colitis induced changes in the gut microbiota profile, including an increase in *Lactobacillus* spp., and amelioration of the symptoms [119]. Similarly, resveratrol (a polyphenol found at high concentrations in grapes) was described to increase Lactobacilli and Bifidobacteria, accompanied by a decrease in inflammation markers in a rat model of colitis [120]. Nevertheless, so far, no study has yet verified the efficacy of prebiotics in patients with CD.

### 3.2. Dietary Interventions Targeting Microbiome in CD

There have been several attempts to identify dietary patterns and the risk of CD progression. A recent meta-analysis identified a “healthy” diet (defined as a high intake of vegetables, fruits, legumes, low-fat dairy products, fiber, poultry, fish, nuts, and whole-grain foods) as a protective factor against CD development [121]. However, the relationship between specific diets and the increased risk of CD is less clear [121]. On the other hand, a relationship between the intake of specific foods and nutrients with a microbiota enriched in bacteria that modulate the inflammatory response has been also proposed [122]. For instance, the consumption of high-sugar foods has been related to the reduced abundance of anti-inflammatory bacteria (*F. prausnitzii* and *Roseburia hominis*), while plant-based foods were linked to an increase in SCFA-producers with potential anti-inflammatory effects [111,122]. Among the proposed mechanisms of action involved in the response to therapy in patients with CD, modulation of the gut microbiota composition by diet appears as one of the most important factors. Thus, the management of CD should not only focus on the use of pharmacological strategies, but it should also include nutritional interventions, especially aimed at modulating the immune response and reversing gut dysbiosis [8,123].

Diet-based therapies targeting the management of CD have been tested since the late 1970s [124]. EEN has become the gold-standard treatment against active CD in pediatric patients affected by luminal CD [8]. EEN has shown to achieve similar outcomes as those obtained with corticosteroid treatment in terms of remission [125], while benefiting bone and muscle parameters, mucosal healing, and growth as well as reducing the risk of relapse in the patients [126]. Such therapeutic effects of EEN have been associated with the exclusion of specific factors from the diet including fats, sugars, or food additives that are likely to be harmful due to their described ability to trigger inflammation [126,127]. Interestingly, studies on the impact of EEN on gut-microbiome dynamics have shown a reduction in α-diversity [128], which has been suggested to favor the long-term restoration of the gut microbiota [129]. However, a recent prospective study carried out by Levine et al. [130], showed a rebound in pre-treatment composition after 12 weeks on EEN. In addition, the lack of palatability of enteral formula commonly leads to difficulties in acceptance and compliance, hindering its implementation on a larger scale [131].

In order to overcome the above-mentioned limitations of EEN, there have been some attempts for seeking alternative nutritional strategies based on whole-food diets for CD treatment. The main objective of these types of diets is to reduce foods that have been described to be pro-inflammatory (e.g., red meat, processed meat, sugar, etc.) [132] and/or increase those types of food that could promote a favorable intestinal microbiota [16,133]. In this sense, the use of partial enteral nutrition (PEN), i.e., consumption of whole food supplemented with enteral nutrition (EN), was introduced as an attempt to improve compliance [134]. The first results obtained from PEN were discouraging as, although positive results were obtained in active CD management, the lack of restriction on the type of whole food consumed led to lower remission rates than EEN [134]. Furthermore, when comparing the effects of both diets, PEN has been described to fail in modulating the gut microbiome [135].

Other diets with anti-inflammatory potential, including the low-FODMAP diet (exclusion of fermentable oligosaccharides, disaccharides, monosaccharides, and polyols) [136,137,138,139], the Specific Carbohydrate Diet (SCD, exclusion of complex carbohydrates) [140,141,142,143], or the Mediterranean Diet [143,144,145,146,147], have been assessed as an alternative to EEN. However, most of these diets have shown contradictory results regarding their effectiveness in CD treatment [148]. The impact of these diets on the pathogenesis and management of CD has already been thoroughly reviewed elsewhere [149,150]. Of special interest is the case of the CD exclusion diet (CDED, low in fat and animal protein with high content of carbohydrates and dietary fiber) [130,151,152,153,154,155], which is emerging as a potent alternative to EEN. It consists of three phases that start with a very restrictive diet supplemented with PEN, which is gradually reduced as new foods are introduced. Clinical trials in pediatric patients with CD have shown that CDED plus PEN has comparable results in inducing remission. Furthermore, the patient’s compliance with an allowance to consume whole foods thereby increases the probability of success [154]. Anyway, all these studies have been focused on clinical outcomes and have disregarded data on the gut microbiome, which is why information on microbial changes associated with such diets is still scarce.

Table 1 summarizes the available data on microbiome and/or microbial metabolism modulation provoked by the different diet-based therapies against CD. In general, one of the common features is the modulation of microbial diversity as the first response to nutritional intervention. Thus, the low-FODMAP diet has been reported to reduce Firmicutes including *Clostridium* cluster XIVa and *F. prausnitzii*. On the other hand, the SCD and Mediterranean diet showed an increased diversity with a reduction in Proteobacteria and Bacillaceae abundance, together with a timid increase in Bacteroidetes and *Clostridium* cluster IV and XIVa. Importantly, these changes seemed to remain in the long-term and did not return to baseline composition.

More recently the use of the CDED (with or without PEN) has been demonstrated not only to successfully induce remission and reduce inflammation comparable to those obtained with EEN but also to promote long-term modifications in microbiome profiles, as well as to increase tolerance and compliance in patients with CD [130,152,159]. For instance, the combination of PEN and CDED reduced the abundance of Actinobacteria and Proteobacteria, with an increase in commensal Clostridia after 6 weeks of diet intervention. This modulation of the gut microbiota supports findings observed with EEN intervention. However, while the microbiome of patients on EEN diet has been described to return to baseline profiles upon long-term remission, changes in the microbiota in patients consuming a CDED were maintained after 12 weeks post-treatment [130].

Notwithstanding, therapeutic diets should be thoroughly supervised by an experienced nutritionist, as any nutritional deficiency or imbalance may lead to negative outcomes such as malnutrition or growth delay in children [149].

## 4. Conclusions

Crohn’s disease is a complex, disabling, idiopathic, progressive, and destructive disorder with an unknown etiology. The management and treatment of CD are currently based on pharmacological strategies with high co-morbidities and health burden associated. Thus, the development of more cost-effective mitigation strategies has become a priority. Over the years, there have been several attempts to develop non-pharmacological therapies to ameliorate CD activity, including nutritional approaches specifically designed to reinforce the immune system and reduce intestinal inflammation.

The relationship between the gut microbiome and CD has been largely studied over the last decades in both preclinical and clinical studies. Indeed, dysbiosis has arisen as a major player in the development of functional and inflammatory intestinal disorders including CD. Thus, the development of strategies aimed at the modulation and restoration of a normally functioning microbiome has become a priority. Current evidence based on preclinical studies using nutritional strategies, including the use of pre-/probiotics has shown promising results in experimental IBD models. However, to date, the attempts to translate such results into human subjects with CD have largely failed.

On the other hand, studies have shown that diet can remarkably impact both the composition and functionality of the gut microbiota to maintain a healthy gut. Nevertheless, the precise knowledge of how specific dietary strategies affect host–microbe interactions in IBD, and more specifically in CD, is still insufficient. In addition, the lack of standardization of clinical trials, variability in the design, and failure in compliance entails a major limitation in these types of studies. Thus, the optimization of nutritional interventions that could complement the currently used pharmacological therapies or therapies used to ameliorate CD symptoms is a field of special biomedical interest. In this regard, while the usefulness of microbial-based therapies is still controversial, the implementation of some diets, such as SCD, low-FODMAP, and more recently CDED, are being recognized as potentially interesting for this purpose, although data are still scarce. Thus, more well-designed, adequately powered randomized, controlled clinical trials are needed to confirm the potential benefit of such nutritional interventions, as well as to unravel the key cellular and molecular players and their specific roles in driving potential health benefits.

## Figures and Tables

**Figure 1 ijms-23-08361-f001:**
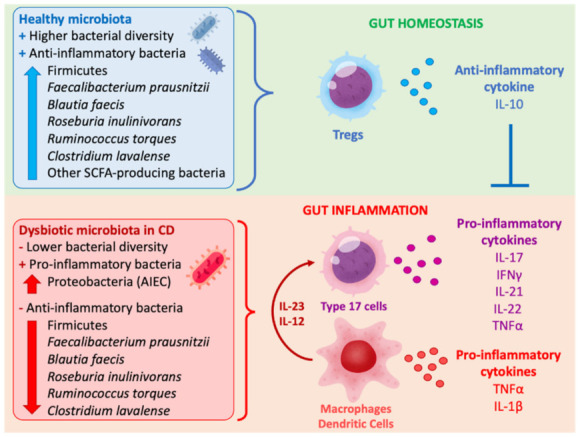
Overview of inflammatory mechanisms involved in CD progression related to gut dysbiosis.

**Table 1 ijms-23-08361-t001:** Clinical trials with nutritional intervention in patients with CD.

Intervention	Cohort and Sample Size	Trial Design and Follow-Up	Objective	Outcomes	Ref
EEN	Pediatric patients with active CD (*n* = 10)	Prospective observational study.	To investigate the impact of EEN therapy on intestinal microbiota in patients with active CD that achieved substantial remission (SR) vs. those that did not achieve SR (non-SR) after 24-weeks follow-up.	↓ α-diversity in SR	[128]
		EEN via nasogastric/gastric tubing for at least 12 weeks to induce remission.		↑ α-diversity in non-SR	
				↓ Firmicute in SR group	
				↑ Bacteroidetes in SR group	
				↓ Bacteroidetes in non-SR group	
				↑ Firmicutes and Verrucomicrobia in non-SR group	
EEN	Pediatric patients with new-onset active CD (*n* = 19)	Randomized, prospective clinical trial. EEN (ModulenÒ IBD, *n* = 13) or corticosteroids (*n* = 6) for 8 weeks.	To investigate differences between EEN vs. corticosteroids on inflammation and intestinal microbiota.	No differences in clinical remission	[156]
		NCT00265772 ^a^		↑ mucosal healing in the EEN group	
				↑ proportion of *Rominococcus* and *Clostridium* in EEN group	
				↓ *Faecalbacterium* and *Roseburia* in EEN group	
				↑ α-diversity in EEN group	
EEN or PEN vs. anti-TNF therapy	Pediatric patients with CD (*n* = 90)	Prospective cohort clinical trial. Consumption of EEN (*n* = 22), PEN (*n* = 16), or treated with anti-TNF therapy (*n* = 52) for 8 weeks.	To evaluate the dynamics of microbiome during treatment.	↓ *Dialister*, *Dorea*, *Gordonibacter*, *Haemophilus* and *Streptococcus* with EEN after 1 week	[135]
				↓ *Candida*, *Clavispora* and *Cyberlindnera* with EEN after 1 week	
				↑ *Alistipes* with EEN after 1 week	
				Microbiota profile closer to healthy controls’ profile (*n* = 26) after 8 weeks of treatment with EEN and anti-TNF	
PEN	Adult patients with active CD (*n* = 17)	Observational study. Daily consumption of E028 (NutriciaÒ) enteral nutrition (*n* = 17) for 2 weeks.	To evaluate changes in microbial metabolism through metabolome analysis and the relation with reduction in inflammation.	↓ CRP	[157]
		07/Q1205/39		↓ SCFA	
				↓ 1-propanol	
				↓ 1-butanol	
				↓ SCFA esters	
PEN	Pediatric patients with CD in clinical remission or mild disease activity (*n* = 41)	Two center, non-randomized controlled intervention study. Daily intake of casein based complete liquid formula (ModulenÒ IBD, *n* = 22) or no nutritional intervention (*n* = 19) for 12 months.	To investigate efficacy of PEN on bone health, growth, and course and assess microbial and metabolome changes.	No differences in bone parameters	[158]
		DRKS00010278		Improved BMI, muscle-cross sectional area and grip strength in PEN group	
				Improved height z-scores in PEN group	
				↑ phosphatidylcholines	
				↑ non-esterified fatty acids	
				↑ fumaric acid	
				↓ α-diversity in PEN group	
Low-FODMAP Diet	Adult patients with quiescent CD (*n* = 9)	Randomized, controlled cross-over, single-blinded clinical trial. Consumption of low-FODMAP diet or a diet containing FODMAP content of a typical Australian diet for 21 days with a 21-day washout period.	To evaluate differences in fecal microbiota, as well as differences in fecal pH, SCFA, GI symptoms, fecal frequency and weight, and whole-gut transit time.	↓ GI symptoms after 14 days in the low FODMAP group	[138]
		ACTRN12612001185853		↓ butyrate-producing *Clostridium* cluster XIVa and mucus-associated *Akkermansia muciniphila* in low FODMAP group	
				↑ *Ruminococcus torques* with low FODMAP diet	
Low-FODMAP Diet	Adult patients with UC or quiescent CD (*n* = 52)	Multicenter, randomized, parallel, single-blinded, placebo-controlled trial. Consumption of low-FODMAP diet (*n* = 27; *n* = 14 with CD) or placebo Sham diet (*n* = 25; *n* = 12 with CD) for 4 weeks.	To evaluate differences in IBS Severity Scoring System, inflammatory markers, and microbiome composition and SCFA.	No differences in SCFA between diets in patients with CD	[137]
		ISRCTN17061468		↓ *Bifidobacterium longun*, *B. adolescentis*, *F. prausnitzii* species in the FODMAP group	
				↑ *B. dentium* in low-FODMAP group	
Specific Carbohydrate Diet	Pediatric patients with mild to moderate IBD (*n* = 12)	Multicenter, open-label clinical trial. Consumption of SCD for 12 weeks (*n* = 9 with CD).	To determine the effect of SCD on active IBD clinical and laboratory parameters as well as in gut microbiome	Improvement in CRP at week 2	[141]
				↓ Calprotectin at week 4	
				↑ Albumin at week 12	
				Improvement of dysbiosis after 2 weeks	
				↑ Inter-individual variability in microbiome dynamics	
Specific Carbohydrate Diet	Pediatric patients with CD	Randomized, double-blind, intervention, controlled clinical trial. Consumption of SCD (*n* = 3), modified SCD (with oats and rice; MSCD, *n* = 4) or whole food diet excluding wheat, corn, sugar, milk and food additives (*n* = 3) for 12 weeks.	To evaluate the efficacy of SCD and two modified versions of SCD on CD clinical parameters and changes gut microbiome.	↑ *Blautia*, *Lachnospiraceae*, *Faecalibacterium prausnitzii*, *Roseburia hominis*, *Roseburia intestinalis*, *Anaerobutyricum hallii* and *Eubacterium eligens*	[140]
	(*n* = 10)	NCT02610101		↓ *Escherichia coli*	
Specific Carbohydrate Diet vs. Low Residue Diet	Adult patients with CD in clinical remission or healthy volunteers (*n* = 8)	Consumption of SCD or LRD for 30 days with a 30-day washout period.	To detect changes in the gut microbiome.	↑ diversity on SCD diet	[142]
				↓ diversity on LRD diet	
Specific Carbohydrate Diet vs. Mediterranean Diet	CD adult patients with mild to moderate symptoms (*n* = 194)	Multicenter, parallel group, randomized controlled trial. Consumption of SCD (*n* = 101) or Mediterranean diet (*n* = 93) for 12 weeks.	To compare the effectiveness of SCD to Mediterranean diet in symptomatic remission of CD.	No differences between diets in CD remission, fecal calprotectin, and CRP after 6 weeks of treatment	[143]
		NCT03058679		No differences in microbiome analysis	
Mediterranean-inspired Diet	Patients with active yet stable CD symptoms	Consumption of Mediterranean-inspired anti-inflammatory diet for 6 weeks.	To evaluate beneficial effects on patients with CD by determining changes in gene expression and microbiota abundance.	Changes in expression of genes involved in EIF2 signaling, B cell development, Th cell differentiation, uracil degradation II and thymine degradation	[144]
	(*n* = 8)	NTY/11/11/109		↑ Bacteroidetes and *Clostridium* cluster IV and XIVa	
				↓ Proteobacteria and Bacillaceae.	
CDED plus PEN	Pediatric patients with mild to moderate luminal CD (*n* = 78)	Multicenter, prospective, randomized controlled trial.	To compare tolerability and efficacy of CDED + PEN with EEN in inducing and sustaining remission.	Higher tolerability to CDED + PEN	[130]
		CDED + PEN (*n* = 40) or EEN (*n* = 34) for 6 weeks.		↓ Actinobacteria and Proteobacteria after 6 weeks with both diets	
		NCT01728870		↑ Clostridia after 6 weeks with both diets.	
				Rebound toward baseline community at week 12 in EEN group	
				Changes in community following the same trend as week 6 at week 12 in CDED + PEN group	

^a^ Registry number of the trial is indicated when available. BMI, body mass index; CD, Crohn’s disease; CDED, Crohn’s disease exclusion diet; CRP, C-reactive protein; EEN, exclusive enteral nutrition; EIF2, eukaryotic initiation factor 2; FODMAP, fermentable, disaccharides, monosaccharides, and polyols; GI, gastrointestinal; IBD, inflammatory bowel disease; LRD, low-residue diet; MSCD, modified specific carbohydrate diet; PEN, partial enteral nutrition; SCD, Specific Carbohydrate Diet; SCFA, short-chain fatty acids; TNF, tumor necrosis factor; UC, ulcerative colitis.

## Data Availability

Not applicable.

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
