# Peer review of "Crohn’s Disease, Host–Microbiota Interactions, and Immunonutrition: Dietary Strategies Targeting Gut Microbiome as Novel Therapeutic Approaches"

_ijms, 2022, doi:10.3390/ijms23158361_

Round 1

Reviewer 1 Report

The focus on dietary strategies is central to the developing therapy of Crohn's disease. Lines 1 through 3 of the abstract identify the authors lack of knowledge of the pathogenesis of Crohn's disease and  current dietary exclusion  therapy.
Crohn's disease is an immune-mediated disease whose immunological dysfunction is the consequence of meaningful microbial challenge in the relative absence of acquired immunity by Mycobacterium avium subspecies paratuberculosis (MAP). Fixation of the immune system's pro-inflammatory response to MAP is now documented.
Current therapy focuses on elimination from diet of foods that have the potential of having been adulterated by MAP and on aggressive management of submucosal polymicrobial infection.
Diet does modulate regional representation of the gastrointestinal microbiota. This ha relevance when it comes attempting to destroy the MAP template through dietary immune system enhancement.

A future starting point for the authors is line 4 and then  they need to do a great deal of reading of published journal articles before attempting to compose  a valid journal commentary.

Author Response

Response to reviewer’s comments: We thank the reviewer for taking the time to revise our manuscript. We agree that Mycobacterium avium subspecies paratuberculosis may be a relevant zoonotic menace and, as many other bacteria currently under study such as Enterococcus faecalis, it might play a role in the establishment of Crohn’s disease by triggering an altered immune response, but its actual role in the development of this disease is still under analysis as contradictory reports have been published (See Kaczmarkowska, A. et al., 2021. DOI: 10.26444/aaem/136398). On the other hand, in this manuscript we have focused on the different currently used by clinicians to manage CD and the potential role of the whole human microbiota in the course of the disease. Furthermore, among the authors of this manuscript are included a clinical endocrinologist, an specialized nurse and two nutritionists with expertise in the management of CD.

Nonetheless, we appreciate the reviewer’s comments, and we will take his/her suggestions into account for future studies and review manuscripts.

Reviewer 2 Report

The manuscript covers a very interesting topic. However, some points need to be clarified and discussed. Here are some general suggestions:

 The text does not completely reflect the title as the part on Dietary interventions targeting microbiome in CD is not as thorough as the first part. 

Furthermore, some points need to be emphasized. The role and efficacy of EEN in the treatment of luminal Crohn's disease (CD) has been well established and is the treatment of choice for patients with pediatric Crohn's disease (it should be mentioned in the introduction where treatment options are discussed). It has also been shown that CDED with partial enteral nutrition has comparable efficacy to EEN therapy alone in inducing remission in children with CD. In contrast, other diets mentioned in the manuscript are not effective in treating Crohn's disease. Therefore, it would be interesting to discuss Table 1 and to compare EEN and CDED with other diets and possibly children vs adults. 

In addition, it has to be mentioned that any exclusion diet must be under control of nutritionst because it may cause nutritional imbalances and deficiencies and in children it can affect growth. 

Author Response

The manuscript covers a very interesting topic. However, some points need to be clarified and discussed. Here are some general suggestions:

Comment 1: The text does not completely reflect the title as the part on Dietary interventions targeting microbiome in CD is not as thorough as the first part. 

Response to reviewer’s comment 1: We thank the reviewer for his/her thoroughly revision of the manuscript and the suggestions. We agree that the title can be confusing, so we have modified it for "Crohn's disease, host-microbiota interactions, and immunonutrition: dietary strategies targeting the gut microbiome as novel therapeutic approaches".

Comment 2: Furthermore, some points need to be emphasized. The role and efficacy of EEN in the treatment of luminal Crohn's disease (CD) has been well established and is the treatment of choice for patients with pediatric Crohn's disease (it should be mentioned in the introduction where treatment options are discussed). 

Response to reviewer’s comment 2: We thank the reviewer for highlighting this relevant issue. We agree that it is important to emphasize the role and efficacy of EEN in the treatment of CD and include it in the introduction. Thus, we have now added a paragraph discussing it (lines 66-74, page 2).

Comment 3: It has also been shown that CDED with partial enteral nutrition has comparable efficacy to EEN therapy alone in inducing remission in children with CD. In contrast, other diets mentioned in the manuscript are not effective in treating Crohn’s disease. Therefore, it would be interesting to discuss Table 1 and to compare EEN and CDED with other diets and possibly children vs adults. 

Response to reviewer’s comment 3: Thanks for the comment. As suggested by the reviewer, we have extended the paragraph regarding the efficacy of other diets and CDED in CD treatment. As highlighted in the text, the effectiveness of each treatment has been recently reviewed by Fitzpatrick et al., 2022 and we have also included a review from Starz et al., 2021 where most of these issues are discussed (lines 442-462, page 10). On the other hand, there are not enough studies to provide a reliable comparison between children vs adults in this review.

Comment 4: In addition, it has to be mentioned that any exclusion diet must be under control of nutritionst because it may cause nutritional imbalances and deficiencies and in children it can affect growth. 

Response to reviewer’s comment 4: We really appreciate this remark. We completely agree with the reviewer that nutritionist control is fundamental for supervising any exclusion diet. Indeed, two of the authors of this manuscript are nutritionists with expertise in the management of these type of patients. We have now amended the text to reflect this issue (lines 499-501, page 13).

Reviewer 3 Report

The manuscript by Núñez-Sánchez et al. is an excellent review of the gut microbiota and Crohn's disease and the effect of dietary strategies on CD.

As noted above, this is an excellent review of the gut microbiota and IBD. It is easily understood and contains excellent references. Therefore, the primary suggestion that I have is to change the title of the manuscript so that it also reflects this aspect of the manuscript and allow searches for host-microbiota interactions in CD to identify this manuscript. Possibly something like "Crohn's disease, host-microbiota interactions, and immunonutrition: dietary strategies targeting the gut microbiome as novel therapeutic approaches".

A few sentences need to be rewritten. Some examples are:

1. "The pathogenesis of CD is multifactorial and involves the interplay between host genetics, immune system, and environmental factors resulting in an aberrant response and subsequent intestinal inflammation." It is not clear what the aberrant response is. Perhaps it refers to aberrant interaction between the microbiome and the host.

2. Similarly, "The pathogenesis of CD is multifactorial and involves the interplay between host genetics, immune system, and gut microbiota, which are influenced by environmental factors and results in an aberrant response and subsequent intestinal inflammation [4]" needs to be clarified.

3. "However, how specific dietary strategies could modulate the gut microbiota composition and the impact on host-microbe interactions in CD is still unclear. In this review, we discuss the most recent knowledge on host-microbe interactions and their involvement in CD pathogenesis and severity, and we highlight the most up-to-date information on gut microbiota modulation through nutritional strategies, focusing on its role in gut inflammation and immunity." This passage should be rewritten. For example:

"However, how specific dietary strategies could modulate the gut microbiota composition and how this would impact host-microbe interactions in CD patients is still unclear. In this review, we discuss the most recent knowledge on host-microbe interactions and its involvement in CD pathogenesis and severity, and we highlight the most up-to-date information on gut microbiota modulation through nutritional strategies, focusing on the role of the microbiota in gut inflammation and immunity."

4. "Gut resident macrophages, together with dendritic cells, play a key role in the establishment of this exacerbated pro-inflammatory process found in CD not only because they are the main producers of IL-23 [21], which activates the Th17 and Th1 inflammatory response, but also in the progression of the disease as a later cellular source of the other pro-inflammatory cytokines, such as TNFα and IL-1β." This sentence should be rewritten. For example:

"Gut resident macrophages, together with dendritic cells, play a key role in the establishment of this exacerbated pro-inflammatory process found in CD not only because they are the main producers of IL-23 [21], which activates the Th17 and Th1 inflammatory response, but they are also involved in the progression of the disease as a later cellular source of other pro-inflammatory cytokines, such as TNFα and IL-1β."

5. "As described above, compositional changes in the gut microbiota are potential contributing factors in driving inflammation in CD. Thus, any alteration in gut bacteria will consequently alter the bacterially generated metabolite landscape of the gut." This passage needs to be rewritten. For example:

"As described above, compositional changes in the gut microbiota are potential contributing factors in driving inflammation in CD. Importantly, alterations in the gut bacteria will alter the bacterially generated metabolite landscape of the gut."

There are also some minor errors in English grammar. The English speaking authors of the manuscript could go over the manuscript and correct these errors.

Author Response

Comment 1: The manuscript by Núñez-Sánchez et al. is an excellent review of the gut microbiota and Crohn's disease and the effect of dietary strategies on CD.

Response to reviewer’s comment 1: Thank you very much for taking the time to review our manuscript and for your kind comment.

Comment 2: As noted above, this is an excellent review of the gut microbiota and IBD. It is easily understood and contains excellent references. Therefore, the primary suggestion that I have is to change the title of the manuscript so that it also reflects this aspect of the manuscript and allow searches for host-microbiota interactions in CD to identify this manuscript. Possibly something like "Crohn's disease, host-microbiota interactions, and immunonutrition: dietary strategies targeting the gut microbiome as novel therapeutic approaches".

 A few sentences need to be rewritten. Some examples are:

  1. "The pathogenesis of CD is multifactorial and involves the interplay between host genetics, immune system, and environmental factors resulting in an aberrant response and subsequent intestinal inflammation." It is not clear what the aberrant response is. Perhaps it refers to aberrant interaction between the microbiome and the host.

  1. Similarly, "The pathogenesis of CD is multifactorial and involves the interplay between host genetics, immune system, and gut microbiota, which are influenced by environmental factors and results in an aberrant response and subsequent intestinal inflammation [4]" needs to be clarified.

  1. "However, how specific dietary strategies could modulate the gut microbiota composition and the impact on host-microbe interactions in CD is still unclear. In this review, we discuss the most recent knowledge on host-microbe interactions and their involvement in CD pathogenesis and severity, and we highlight the most up-to-date information on gut microbiota modulation through nutritional strategies, focusing on its role in gut inflammation and immunity." This passage should be rewritten. For example:

"However, how specific dietary strategies could modulate the gut microbiota composition and how this would impact host-microbe interactions in CD patients is still unclear. In this review, we discuss the most recent knowledge on host-microbe interactions and its involvement in CD pathogenesis and severity, and we highlight the most up-to-date information on gut microbiota modulation through nutritional strategies, focusing on the role of the microbiota in gut inflammation and immunity."

  1. "Gut resident macrophages, together with dendritic cells, play a key role in the establishment of this exacerbated pro-inflammatory process found in CD not only because they are the main producers of IL-23 [21], which activates the Th17 and Th1 inflammatory response, but also in the progression of the disease as a later cellular source of the other pro-inflammatory cytokines, such as TNFα and IL-1β." This sentence should be rewritten. For example:

"Gut resident macrophages, together with dendritic cells, play a key role in the establishment of this exacerbated pro-inflammatory process found in CD not only because they are the main producers of IL-23 [21], which activates the Th17 and Th1 inflammatory response, but they are also involved in the progression of the disease as a later cellular source of other pro-inflammatory cytokines, such as TNFα and IL-1β."

  1. "As described above, compositional changes in the gut microbiota are potential contributing factors in driving inflammation in CD. Thus, any alteration in gut bacteria will consequently alter the bacterially generated metabolite landscape of the gut." This passage needs to be rewritten. For example:

"As described above, compositional changes in the gut microbiota are potential contributing factors in driving inflammation in CD. Importantly, alterations in the gut bacteria will alter the bacterially generated metabolite landscape of the gut."

Response to reviewer’s comment 2: We truly thank the reviewer for his/her kind comments and helpful suggestions. We have changed all the indicated sentences as suggested throughout the text.

There are also some minor errors in English grammar. The English speaking authors of the manuscript could go over the manuscript and correct these errors.

 Response to reviewer’s final comment: Following reviewer’s suggestion the text has been thoroughly reviewed by an English speaking author.

Round 2

Reviewer 2 Report

I would like to thank the authors for answering all the comments and for correcting the manuscript according to suggestions.